

# Identification and characterization of cold-responsive aquaporins from the larvae of a crambid pest *Agriphila aeneociliella* (Eversmann) (Lepidoptera: Crambidae)

Chunqing Zhao[1], Zhen Liu[2], Yong Liu[1] and Yidi Zhan[1]

[1] College of Plant Protection, Shandong Agricultural University, Taian, China
[2] Weihai Huancui District Bureau of Agriculture and Rural Affairs, Weihai, China

## ABSTRACT

As small ectotherms, insects need to cope with the challenges of winter cold by regulating the water content through water transport. Aquaporins (AQPs) are key players to enhance the cold resistance by mediating essential homeostatic processes in many animals but remain poorly characterized in insects. *Agriphila aeneociliella* is a newly discovered winter wheat pest in China, and its early-stage larvae have strong tolerance to low temperature stress. Six AQP genes were identified, which belong to five AQP subfamilies (RPIP, Eglp, AQP12L, PRIP, DRIP). All of them contained six hydrophobic transmembrane helices (TMHs) and two relatively conservative Asparagine-Proline-Alanine motifs. The three-dimensional homology modeling showed that the six TMHs folded into an hourglass-like shape, and the imperceptible replace of four ar/R residues in contraction region had critical effects on changing the pore size of channels. Moreover, the transcript levels of AaAQP 1, 3, and 6 increased significantly with the treatment time below 0 °C. Combined with the results of pore radius variation, it is suggested that AaAQP1 and AaAQP3 may be considered to be the key anti-hypothermia proteins in *A. aeneociliella* by regulating rapid cell dehydration and allowing the influx of extracellular cold resistance molecules, thus avoiding death in winter.

## INTRODUCTION

At low temperature many insects lose extracellular ion homeostasis and the capacity to mitigate homeostatic imbalance determines their cold tolerance (*Overgaard, Gerber & Andersen, 2021*). The content of free water and antifreeze substances is often closely related to this physiology-based work (*Clark & Worland, 2008*). As small ectotherms, insects need to cope with the challenges of winter cold by regulating the water content and/or the concentration of cold resistance molecules through water transport (*Teets, Marshall & Reynolds, 2023*). For most insects, when the temperature drops below the supercooling point (SCP), ice crystals rapidly form and expand in the cells. As the cells dehydrate, an

Corresponding author
Yidi Zhan, zhanyidi@sdau.edu.cn

osmotic pressure builds between the intracellular and extracellular, causing serious damage to the cells, which are the important reasons leading to their death (*Koop & Zobrist, 2009*).

Insects exposed to cold conditions during winter have developed different behavioural, physiological and morphological adaptations (*Kostál, Vambera & Bastl, 2004*; *MacMillan et al., 2012*). These adaptations could improve the ability to reduce ice crystal formation and maintain osmotic homeostasis (*Toxopeus et al., 2019*), for example, by producing low molecular weight metabolites (*Matsukura et al., 2008*; *Chen, Liang & Zou, 2010*), ice-nucleating agents (*Koop & Zobrist, 2009*), lipids (*Toxopeus et al., 2019*) and aquaporins (*Cohen, 2012*; *Ekert et al., 2016*).

As water channels, aquaporins (AQPs) mediate essential homeostatic processes including water balance (*Tsujimoto et al., 2023*). AQPs are proteins located at the cell membrane to transport water and other small solutes across cell membranes (*Cohen, 2012*; *Abascal, Irisarri & Zardoya, 2014*; *Tsujimoto et al., 2023*). Studies show that AQPs could be used by many insects to against environmental stresses, such as low temperatures and drought (*Luo et al., 2022*), and insects could develop resistance to insecticides through increasing excretion rate by AQPs (*Tan & Chen, 2023*). When insect is in a hypothermia or dehydration environment, AQPs can regulate cell volume and permeability effectively (*Ekert et al., 2016*). At the same time, the influx of cryoprotectants helps to avoid intracellular ice formation (*Toxopeus & Sinclair, 2018*). AQPs only allow water pass through cell membrane, while aquaglyceroporins have a much higher permeability to neutral molecules than water, such as *Escherichia coli* glycerol facilitators which transport glycerol and other linear polyalcohols across cellular membranes (*Fu et al., 2000*).

In arthropods, AQPs are classified into six major clades based on phylogenetic analyses: *Pyrocoelia rufa* integral proteins (PRIP) for transporting water or urea, water-specific *Drosophila* intrinsic protein (DRIP), water and glycerol transporting aquaglyceroporin (RPIP), glycerol-permeating entomoglyceroporin (Eglp), water-impermeable but cation-permeable big brain proteins (BIB), and unorthodox aquaporin (AQP12L) (*Finn et al., 2015*; *Wei, Zhao & Yi, 2017*). Glycerol conductance can be important for survival, as some arthropods accumulate glycerol and other polyols to survive cold or desiccating conditions (*Tsujimoto, Sakamoto & Rasgon, 2017*). Drip and Prip are water-selective, and they mainly transport water molecules (*Tan & Chen, 2023*). They play crucial roles in the diuresis of blood-feeding and phloem-sucking insects (*Esquivel, Cassone & Piermarini, 2016*). AaAQP5 of *Aedes aegypti* participates in transcellular water transport across the Malpighian tubules of larvae where global AaAQP5 expression is important for larval survival (*Misyura, Yerushalmi & Donini, 2017*). The functional units of AQPs exist as a tetramer on the cell membrane, and each monomer of the tetramer function as an independent water channel (*Sui et al., 2001*). Each monomer consists of a peptide chain which typically contains six tandem transmembrane helices (TMHs) and two highly conserved Asparagine-Proline-Alanine (NPA) motifs (*Fu et al., 2000*). The TMHs are linked by five loops, which form an hourglass structure perpendicular to the cell membrane, and the COOH- and $NH_2$-terminal of the TMHs are located on the cytoplasmic side (*Sui et al., 2001*; *Gonen & Walz, 2006*; *Van Ekert et al., 2016*). Two NPA motifs constitute the important functional structure of AQPs. The peptide chain stacking

causes the two NPA to align together to form a narrow pore that controls water flow (*Wallace et al., 2012*). The transport selectivity of AQPs is determined by the narrow "aromatic/arginine" (ar/R) selective filter which act as an electrostatic proton barrier, regulating the transport selectivity according to the size and hydrophobicity of solutes. AQPs and aquaglyceroporins differ in pore size and channel amino acid composition (*Stavang et al., 2015*).

*Agriphila aeneociliella* (Eversmann, 1844) (Lepidoptera: Crambidae) infests on gramineous crops, which mainly distribute in northern hemisphere (*Shodotova, 2008*; *Chi et al., 2016*; *Zhan et al., 2023*). Its early-stage larvae have a strong adaptedness to low temperature stress and can feed on wheat tillers below the land surface 1–2 cm during winter (*Zhang et al., 2015*; *Zhan, Liu & Liu, 2020*). Our previous research has shown that the SCP of the non-diapause *A. aeneociliella* larvae can reach approximately −23 °C, which is rare in the superfamily Pyraloidea. Moreover, compared to other insects of Pyraloidea, the larvae of *A. aeneociliella* are easier to adapt to prolonged period of low temperature exposure (*Zhang et al., 2015*). In order to explore the functional significance of AQPs in the extremely cold tolerance of *A. aeneociliella* larvae, the present study screened and identified AQP genes, analyzed the three-dimensional structure and sequence structure of AQPs, and compared the gene expression levels under different low temperature stress. The results could provide insights for the mechanism of the larval outbreak of *A. aeneociliella* in the winter wheat fields, and also for developing a strategy to control *A. aeneociliella* by the utilization of regulated mechanisms of their AQPs.

## MATERIALS AND METHODS

### Insects

The larvae of *A. aeneociliella* were collected from wheat fields of Shandong Province, China (E 120°21, N 36°46), and were reared individually in a 30 mL perforated plastic box in laboratory. The larvae were reared to the third generation in the artificial climate chamber at 23 ± 1 °C, 14 h: 10 h L: D photoperiod, 70% RH (*Zhang et al., 2015*).

### Treatment design and sample preparation

Based on the data from the Moderate Resolution Imaging Spectroradiometer (MODIS) onboard NASA's Terra and Aqua spacecraft, the daytime land surface temperature (LST) in winter in Shandong Province, China ranges from 0 °C to 10 °C, and the nighttime LST from −8 °C to −2 °C (*Chen, 2021*). Therefore, four temperatures of 4 °C, 0 °C, −4 °C, and −8 °C were chosen as different treatments in the experiment to compare the expression levels of *AQPs*. Thirty third instar larvae with similar body size were selected and transferred into one petri dishes (diameter = 12 cm) sealed with plastic wrap. Then, the larvae were exposed to one of four temperatures (4 °C, 0 °C, −4 °C, −8 °C) for 3 h, 12 h and 24 h in climate chamber. Each treatment was conducted with three biological replicates. The larvae reared at 23 °C for 3 h, 12 h and 24 h in climate chamber were set as control groups. The treated larvae were immediately collected and immersed in liquid nitrogen and stored at −80 °C until utilized for RNA extraction.
## Total RNA extraction and cDNA synthesis

Thirty third instar larvae of *A. aeneociliella* were grinded into powder in 1.5 mL tubes placed in liquid nitrogen. Total RNA was extracted using TRIzol reagent (TransGen Biotech, Beijing, CHN) following manufacturer's instructions. The integrity of RNA was verified by 1.5% agarose gel electrophoresis. The quality of total RNA was then detected using ultraviolet spectrophotometry (NanoDrop One; Thermo Fisher, Waltham, MA, USA) (OD260/OD280 = 1.92, OD260/OD230 = 2.0). First strand cDNA was reverse transcribed from RNA using PrimeScript II Rtase with the Oligo (dT) primer and Random 6 primer (PrimeScript™ II 1st Strand cDNA Synthesis Kit, Takara Bio, Beijing, China). The cDNA was stored at −20 °C for further experiment.

## DNA amplification, cloning and sequence analysis

Specific primers of *AQPs* sequences were designed by Primer premier 5.0 based on cloned sequence data of *A. aeneociliella* and all of the primers used in this study were listed in Table S1. The PCR was carried out in Bio Red T100™ Thermal Cycler by using 2× Accurate Taq Master Mix (Accurate Biology), with the following cycling parameters: initial denaturation for 3 min at 98 °C, followed by 35 cycles of 10 s at 98 °C, 30 s at 45–72 °C, 1 min at 72 °C, and a final extension for 10 min at 72 °C. The PCR products were gel-purified by using Gel Extraction Kit (Beijing CoWin Biotech Co,. Ltd, Beijing, CHN), and inserted into pMD18-T vector (Takara Bio, Beijing, CHN). The recombinant plasmids were transformed into competent *E. coli* cells (DH5α) for Sanger sequencing by TSINGKE, Qingdao, China.

The DNA sequences were analyzed using the Blastx search program (http://blast.ncbi.nlm.nih.gov/Blast.cgi) and aligned with AQPs from other insect species by using ClustalW. Physicochemical property (pI/MW and Hydrophobicity) of the AQP proteins were predicted using ExPASy tools (http://us.expasy.org/tools/). Transmembrane domains were predicted with TMHMM (https://services.healthtech.dtu.dk/services/TMHMM-2.0/). Functions of the AQP proteins were predicted by InterPro (http://www.ebi.ac.uk/interpro/). Phylogenetic analysis of six *A. aeneociliella* AQPs and 47 other arthropods AQPs downloaded from GenBank were performed using maximum likelihood method in MEGA 7.

## Molecular modeling

Three-dimensional constructions of AQPs were generated by using SWISS-MODEL (https://swissmodel.expasy.org/) for homology modeling. The crystal structure of AQP1 ortholog (PDB ID 1J4N) and human AQP5 (PDB ID 1LDA) were chosen as the best template because of their high resolution (1.8 Å) and the high level of amino acid sequence identity (32–40%) to the AaAQPs. Briefly, six AaAQP sequences were aligned by using the Phyre2 tool. Pymol was used to visualize the three-dimensional coordinates for the atoms of the model. Pore diameters of homology models were calculated using the HOLE2.0 program of the energy minimized homology model structure using the simple rad van der Waals radius file.

### Real-time qPCR analysis

Specific RT-qPCR primers were designed using Primer Premier 5.0 (Table S1). The RT-qPCR analysis was conducted using CFX-96 real-time PCR Detection System (Bio-Rad, CA, USA). First-strand cDNA was used as the templates to examine the relative expression of AQP genes in a 20 µL reaction system containing 3 µL of cDNA, 0.8 µL each of 10 µmol/L forward primer and reverse primer, 10 µL of 2× T5 Fast qPCR Mix (TSINGKE, China), and 5.4 µL of nuclease-free water. Two-step thermocycling conditions were used as follows: 95 °C for 1 min, followed by 40 cycles of 95 °C for 10 s, insert melting curve analysis at 60 °C to 65 °C in increments of 0.55 °C for 15 s. *β-actin* (accession number: OR526988) was used as reference gene. Three biological replicates were performed, and each biological replicate was conducted with three technical replicates.

### Statistical analyses

Relative expression level of *AQPs* was determined using the $2^{-\Delta\Delta Ct}$ method (*Livak & Schmittgen, 2001*), $\Delta\Delta Ct = \Delta Ct$ (cold treatment)—$\Delta Ct$ (control group). All data obtained from the RT-qPCR were analyzed using analysis of ANOVA and then the results were adjusted using a least-significant different (LSD) multiple comparison test. The statistical analyses were conducted using SPSS 19.0 for windows.

## RESULTS

### Identification of AQPs from *A. aeneociliella*

Six *AaAQP* genes were identified in *A. aeneociliella* (Table 1). Six *AaAQPs* shared 57% to 93% identity with amino acid sequences of AQPs from other insect species. Aa*AQP6* of *A. aeneociliella* showed the highest similarity with AQP12L2 from the striped stem borer *Chilo suppressalis* (Walker) (93%). *AaAQP3* shares the lowest identity (57%) with the wax moth *Galleria mellonella L.* The results indicated that AQPs are relatively conserved in Lepidoptera. The encoded protein length of *AaAQP* genes range from 251 to 292 aa, with their molecular weight (MW) from 26.006 to 32.170 kDa and the isoelectric points (PI) from 5.27 to 8.52 (Table 1).

### Sequence analysis of AQPs

As presented in Fig. 1, AaAQPs showed relatively conservative motifs and universal molecular structures in sequence alignment. Each AQP sequence contained six hydrophobic TMHs, which were rich in α-helix, but lack of β-pleated sheet. There were some slightly differences in the family signature motifs between six AQP sequences. AaAQP1, AaAQP2 and AaAQP5 have two conserved NPA (Asn-Pro-Ala) boxes, which are the most common hallmarks among aquaporin family. AaAQP3 and AaAQP4 only have one NPA motif and one NPF motif (Asn-Pro-Phe). While AaAQP6 has the conserved CPY (Cys-Pro-Tyr) and NPV (Asn-Pro-Val) motifs. Residues (Phe, His, Ala, Ser, Asn or Arg) that comprised the ar/R selectivity filter were found in AaAQP sequences except AQP6, the site of Phe[63] and Arg[201] exhibited a highly conservation especially. In the amino acid sequence, all six AQPs sequences had phosphorylation site and N-myristoylation but did not have O-glycosylation. The NISQ and NCSV sites locate at

**Table 1 Blast matches of *AaAQPs*.**

| Gene name | Accession No. | ORF length (nt/aa) | Molecular weight (KDa) | Isoelectric point | Best blastx match | Best blastx match Acc.No. | Identity (%) |
|---|---|---|---|---|---|---|---|
| *AaAQP1* | MW543947 | 807/268 | 27.038 | 6.18 | aquaporin AQPAe.a (*Ostrinia furnacalis*) | XP_028159475.1 | 89.96 |
| *AaAQP2* | MW543948 | 777/258 | 26.890 | 6.88 | aquaporin-1 variant A (*Chilo suppressalis*) | AFC34081.1 | 87.60 |
| *AaAQP3* | MW543949 | 759/252 | 26.899 | 5.27 | aquaporin-4-like (*Galleria mellonella*) | XP_026755324.1 | 57.20 |
| *AaAQP4* | MW543950 | 756/251 | 27.348 | 6.89 | aquaporin-like (*Ostrinia furnacalis*) | XP_028175747.1 | 64.17 |
| *AaAQP5* | MW543951 | 771/256 | 26.006 | 8.52 | aquaporin-like (*Trichoplusia ni*) | XP_026747737.1 | 60.70 |
| *AaAQP6* | MW543952 | 879/292 | 32.170 | 5.43 | Aquaporin AQP12L2 (*Chilo suppressalis*) | AWT57818.1 | 93.56 |

the position 44 of AQP1 and position 165 of AQP3, respectively, which were predicted to be the N-glycosylation.

The phylogenetic tree was conducted to compare six *A. aeneociliella* AQPs and 46 arthropod AQPs using maximum likelihood method. As shown in Fig. 2, six AaAQPs of *A. aeneociliella* were distinct classified into five major subfamilies: DRIP, PRIP, RPIP, Eglp and AQP12L, but no BIB-like sequence was identified. The five subfamilies were further classified into four clades (A–D). The A-D clades represented the classical aquaporins, the aquaglyceroporins (Glp), AQP12L and BIBs, respectively. Based on the phylogenetic analysis, AQP1 and AQP2 were water-specific classical aquaporins, AQP3, AQP4 and AQP5 were Glps. The BIBs group was not identified in *A. aeneociliella* AQPs, which mainly responsible for cation transport (Fig. 2).

**Three-dimensional structure of AaAQPs**

A three-dimensional (3D) construction of AQP1 was generated with SWISS-MODEL and Pymol for homology modeling by using the crystallographically resolved AQP1 ortholog (PDB ID 1J4N) as a template. The structure of AQP1 shared the highest homology with that of AaAQP1 (40% identity). Extracellular view of the structure showing that, AQPs usually function as a tetramer unit with each monomer proving an independent water channel (Fig. 3A). Combined with hydrophobicity analysis, the transmembrane region of each channel was hydrophobic, and both sides out of the membrane were hydrophilic (Fig. S1). Water molecules were transported through the hydrophobic membrane channels, establishing a coordinated water transport pathway (Fig. 3B). The 3D models of AaAQP1 monomer comprise six transmembrane (TMD1-6) and two half-transmembrane helices, forming a typical hourglass-shaped structure of AQPs, and each of the half-transmembrane helices contains a conserved NPA motif that predicted to site one side of the constriction pore (Fig. 3C).

Imperceptible changes happened in four ar/R residues of AaAQPs sequences had critical effects on constriction structure and transport characteristics. Representative

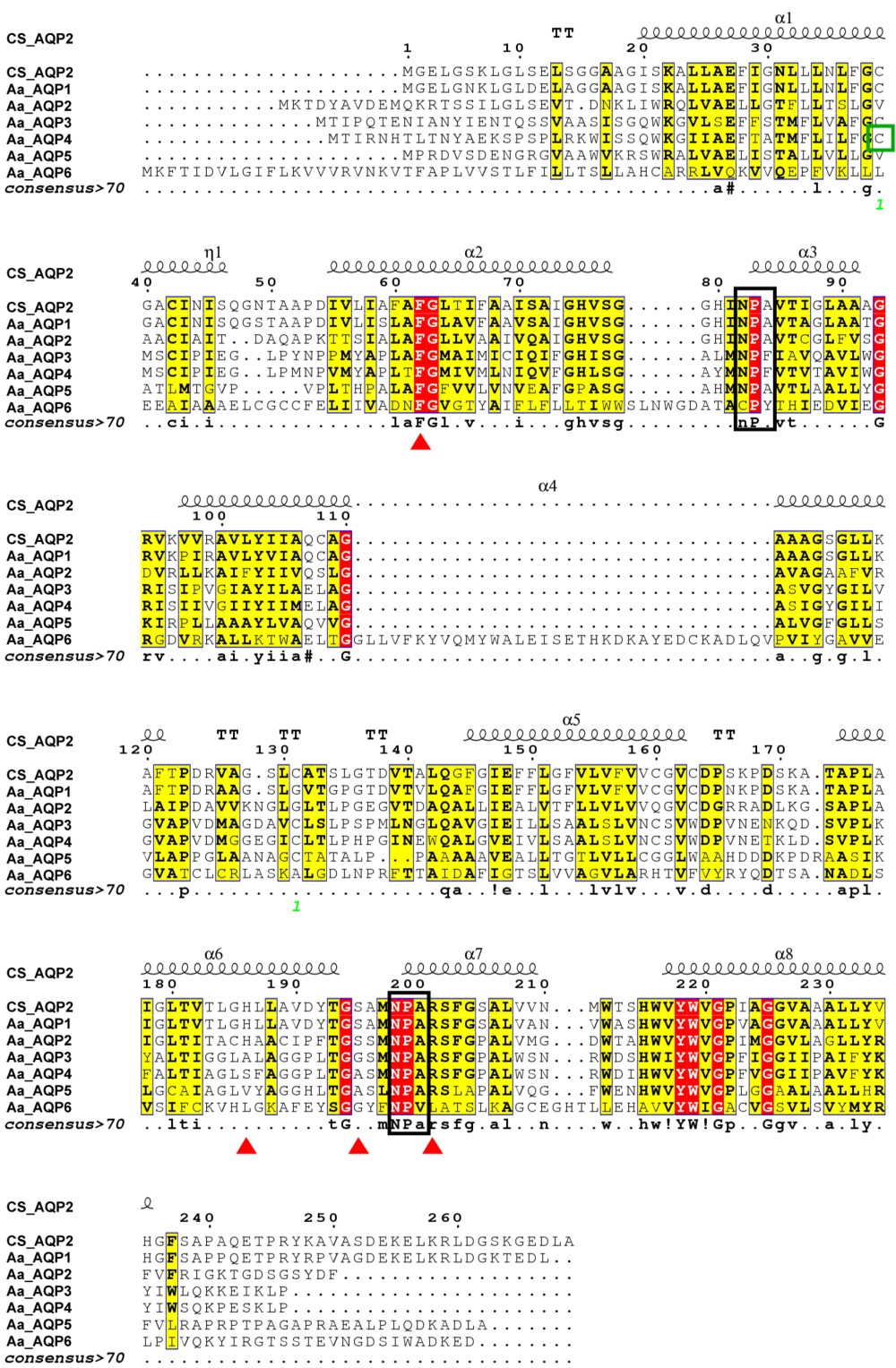

**Figure 1 Multiple amino acid sequence alignment and secondary structure of *Agriphila aeneociliella* aquaporins.** The alignment included AaAQP1-6 and *Chilo suppressalis* AQP2 (AHF53540.1). Two conserved NPA motifs were marked with black boxes and a predicted cysteine residue for mercury-sensitivity with green box. ar/R residues related to osmotic selectivity exhibited a certain variation and were highlighted in red triangles.                 

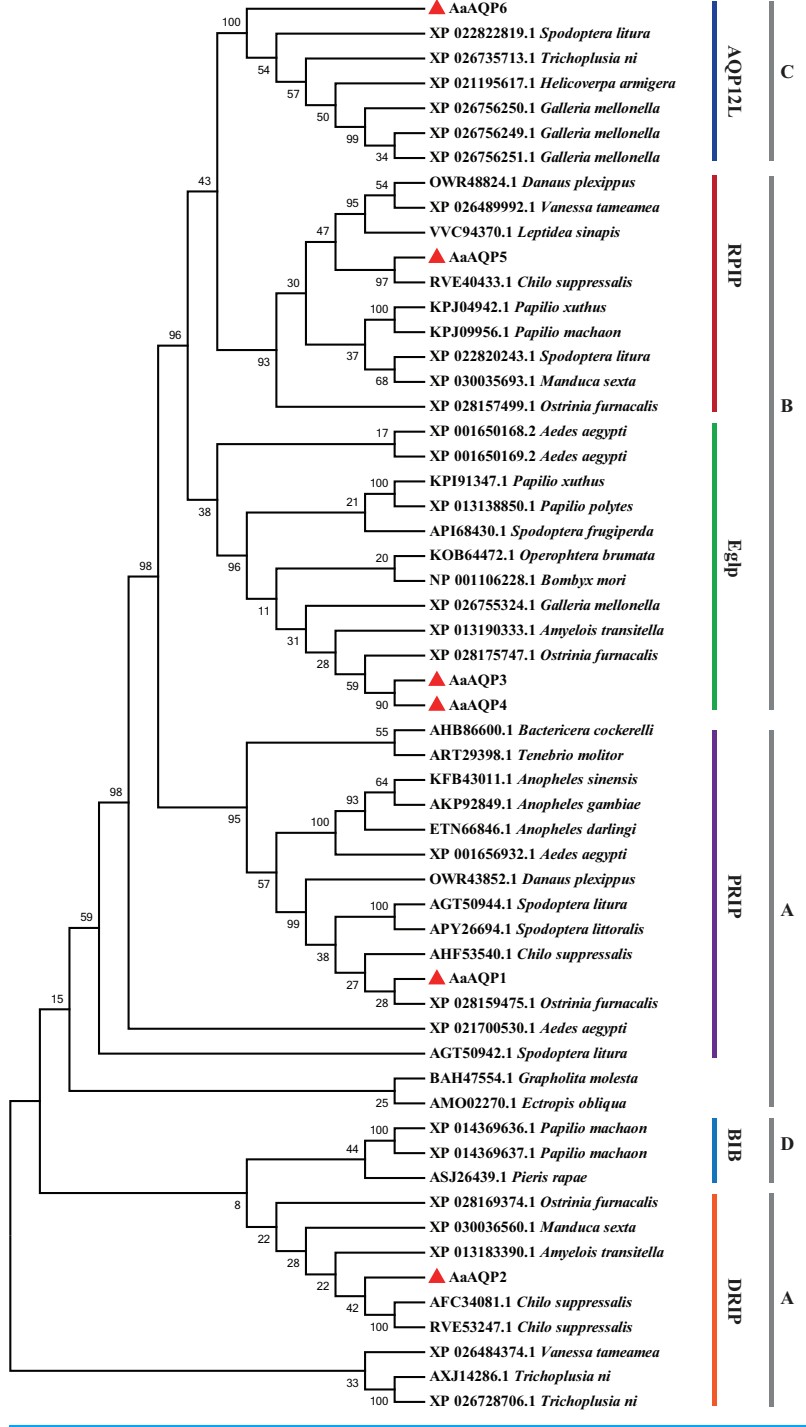

**Figure 2 Phylogenetic analysis and classification of AaAQP1-6 with AQPs from other insects.**
Sequence alignment and phylogenetic analysis of six *Agriphila aeneociliella* AQPs and 46 other
arthropod aquaporins from GenBank by using maximum likelihood method in MEGA 7. The numbers
on the branches represent the bootstrap value inferred from 1,000 sample replicates and the evolutionary
distances were computed using the Poisson correction method. According to *Fabrick et al. (2014)* and
*Wallace et al. (2012)*, representatives from six major arthropod subfamilies (DRIPs, PRIPs, RPIPs, BiBs,
Eglps and AQP12L) are shown, Group A-D represent classical aquaporins, aquaglyceroporins (Glp),
AQP12L and BIBs, respectively.

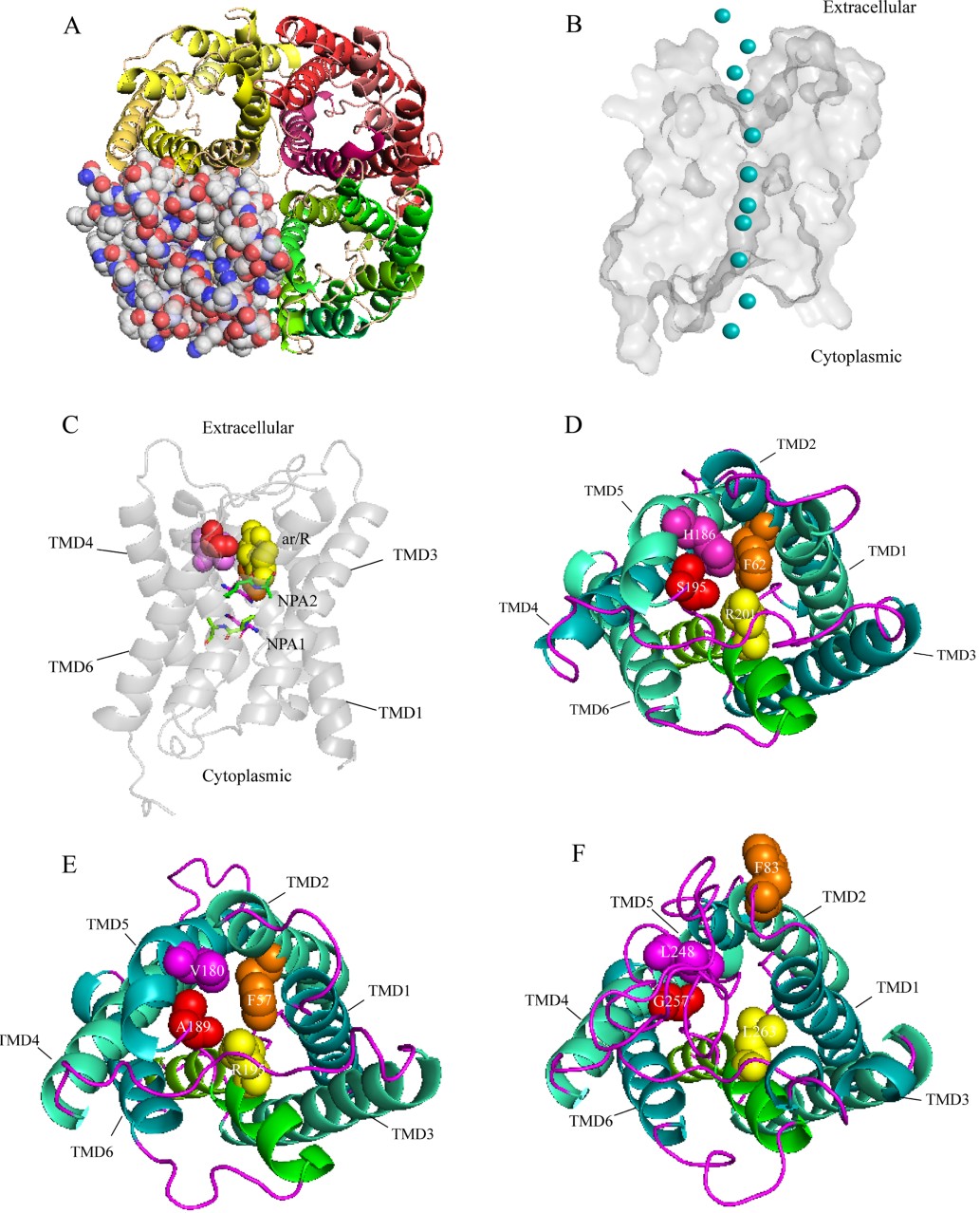

**Figure 3 Structure and diversity of *Agriphila aeneociliella* aquaporins.** (A) Homology modeling of AQP-Agr1 showing extracellular view of the tetramer structure through cartoon (red, green and yellow) and spheres renderings. (B) "Hourglass" model of AQP (side view). Blue circles represent water molecules pass through the narrow channel. (C) Monomeric view in the direction parallel to the membrane of AQP1 showing the aromatic/arginine (ar/R, spheres) and Asn-Pro-Ala (NPA, sticks) constrictions. (D–F) Extracelluar views of cartoon renders of AQP 1, 3 and 6 with ar/R residues (spheres) and Helices (TMD1-6) are annotated.

sequence (AQP1, AQP3 and AQP6) from each of the A-C groups was selected separately to construct a 3D homology model to observe the spatial distribution of ar/R region (Figs. 3D–3F). Clearly, water-specific classical aquaporins (D) equipped with a tighter pore

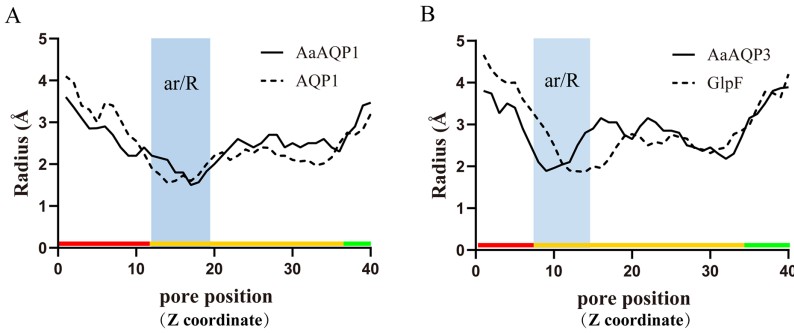

**Figure 4 Pore radius variations along the pore length for different permeating channel.** (A) Comparison of the water-specific channel of AaAQP1 and AQP1 structures. (B) Aquaglyceroporins pore radius of AQP3 and *E.coli* glycerol facilitator (GlpF). The calculated pore radius along the z-coordinate of the pore across is shown. The NPA regions are marked by pink and blue area, respectively. Pore diameters of homology models were calculated using the HOLE2.0 program.

which contraction region radius close to 1.5 Å (Fig. 4), allowing water molecules (diameter = 2.8 Å) to pass through. In more relaxed structure of Glps, the ar/R shrinkage diameter was enlarged to a radius of 1.9 Å through certain amino acid replacement of His[186] and Ser[195], which made it possible to infiltrate glycerol, urea and other molecules.

In order to investigate the relationship between constriction structure and permeability properties of ar/R under freezing stress, an overview of the ar/R residues' composition, corresponding permeability and predicted functions that have been confirmed in previous studies were presented in Table 2. Classical aquaporin subfamily that only allows water pass through exhibited a highly conservative features in constitution, which is responsible for insect cold tolerance by maintaining water homeostasis. However, a diversified residue variation took place in Glps. As the diversity of permeability, Glps allowed cryoprotectant such as glycerol and trehalose to enter the cells, improving the tolerance of insects to low temperature. AQP12L exhibited conservative features in constitution, and is also involved in osmoregulatory processes.

## Expression profiles of *AaAQPs* exposed to low temperatures

All six *AaAQP* genes showed different expression profiles when the larvae were exposed to 4 °C, 0 °C, −4 °C, −8 °C during the same cooling duration. The expression levels of *AQP1* had a significant up-regulation at −8 °C for the time points of 12 h ($P = 0.029$) and 24 h ($P = 0.016$). Similar expression trends were also found in *AQP6* expression, which significantly increased under 0 °C, −4 °C and −8 °C for 24 h (0 °C: $F = 22.170$, $P = 0.02$; −4 °C: $F = 42.350$, $P < 0.001$; −8 °C: $F = 91.591$, $P < 0.001$), and no significant differences were found in response to 3 h cooling time under all temperatures ($F = 1.333$, $P = 0.412$) (Fig. 5). The transcription levels of *AaAQP1*, *AaAQP3*, and *AaAQP6* increased significantly with the extension of treatment time below 0 °C. However, the expressions of *AQP4* ($F = 8.420$, $P = 0.018$) and *AQP5* ($F = 31.523$, $P = 0.001$) decreased significantly when the cooling time extended to 24 h under −8 °C (Fig. 5).

**Table 2** Multiple ar/R residues and corresponding functions of insect aquaporins.

| Gene name | ID number | ar/R | Function under cold stress |
|---|---|---|---|
| **Classical aquaporins** | | | |
| *Grapholita molesta* AQP-Gra1 (*Kataoka, Miyake & Azuma, 2009*) | AB469882 | F H S R | Water-specific transportation |
| *Belgica antarctica* BaAQP1 (*Goto et al., 2011*) | – | F H S R | Water-specific; retain cellular viability during freezing |
| *Spodoptera litura* AQP1 (*Liu et al., 2013*) | KC999953 | F H S R | Responsible for cellular water osmotic pressure regulation |
| *Dendroctonus armandi* DaDrip_v1 (*Fu et al., 2019*) | MH579723 | F H S R | Play key role in cold tolerance by maintaining water homeostasis |
| **Glp** | | | |
| *Bombyx moxi* AQP-Bom2 (*Kataoka, Miyake & Azuma, 2009*) | AB245966 | F S A R | Transport water, glycerol and urea in midgut |
| *Eurosta solidaginis* EsAQP1 (*Philip, Kiss & Lee, 2011*) | FJ489680 | F H A R | Urea and water transporters; promoting freezing tolerance |
| *Lygus Hesperus* LhAQP4A (*Fabrick et al., 2014*) | KF048099 | F H S R | Maintaining water homeostasis |
| *Microdera punctipennis* MpAQP1 (*Li, Liu & Ma, 2017*) | KX129951 | F H A R | Improving cold tolerance by involving in osmoregulation |
| **AQP12L** | | | |
| *Aedes aegypti* AaAQP6 (*Sandhya & Kavitha, 2018*) | KX246910 | F V G L | Play an osmoregulatory function in the pupae |
| *Dendroctonus armandi* DaAqp12L (*Fu et al., 2019*) | MH579727 | F V G L | Maintaining water homeostasis |

# DISCUSSION

An appropriate internal water and cryoprotective compounds management system plays vital role for insects to survival in freezing (*MacMillan et al., 2012*; *Tsujimoto et al., 2023*). AQPs, as conserved transmembrane channel proteins in some insect species, for example, *C. suppressalis* (*Izumi et al., 2006*), *Eurosta solidaginis* (*Philip & Lee, 2010*), and *Dendroctonus armandi* (*Fu et al., 2019*), are mainly responsible for the transport and regulation of water movement in cells and tissues (*Philip & Lee, 2010*; *Toxopeus & Sinclair, 2018*), which playing vital roles in improving insect cold tolerance.

In our study, six *AaAQP* genes were identified from *A. aeneociliella* and their sequence characteristics were also analyzed. The AaAQPs sequences showed appropriate variation in the conserved NPA motifs among different subfamily. Similar changes have also found in other insects, such as *D. armandi* (*Fu et al., 2019*) and *Aedes aegypti* (*Sandhya & Kavitha, 2018*). The function of some AQPs were usually regulated by phosphorylation and glycosylation. The N-terminal residues of aquaporin-1 in *Bemisia tabaci* was rich in phosphorylation sites, inducing *Bt*AQP1 to localize on the cell membrane (*Mathewa et al., 2011*). AQPs lacking glycosylation modification were reported to be localized on the intracellular organelles (*Hendriks et al., 2004*) and lack of the ability to maintain homeostasis (*Fabrick et al., 2014*). Therefore, it can be inferred that there may be some differences between *A. aeneociliella* AQPs in functional regulation and cellular localization. There is usually an $Hg^{2+}$-sensitive cysteine (Fig. 1) in amino acid sequence of AQPs, which can inhibit the process of water permeability reversibly (*Kataoka, Miyake & Azuma, 2009*; *Wei, Zhao & Yi, 2017*).

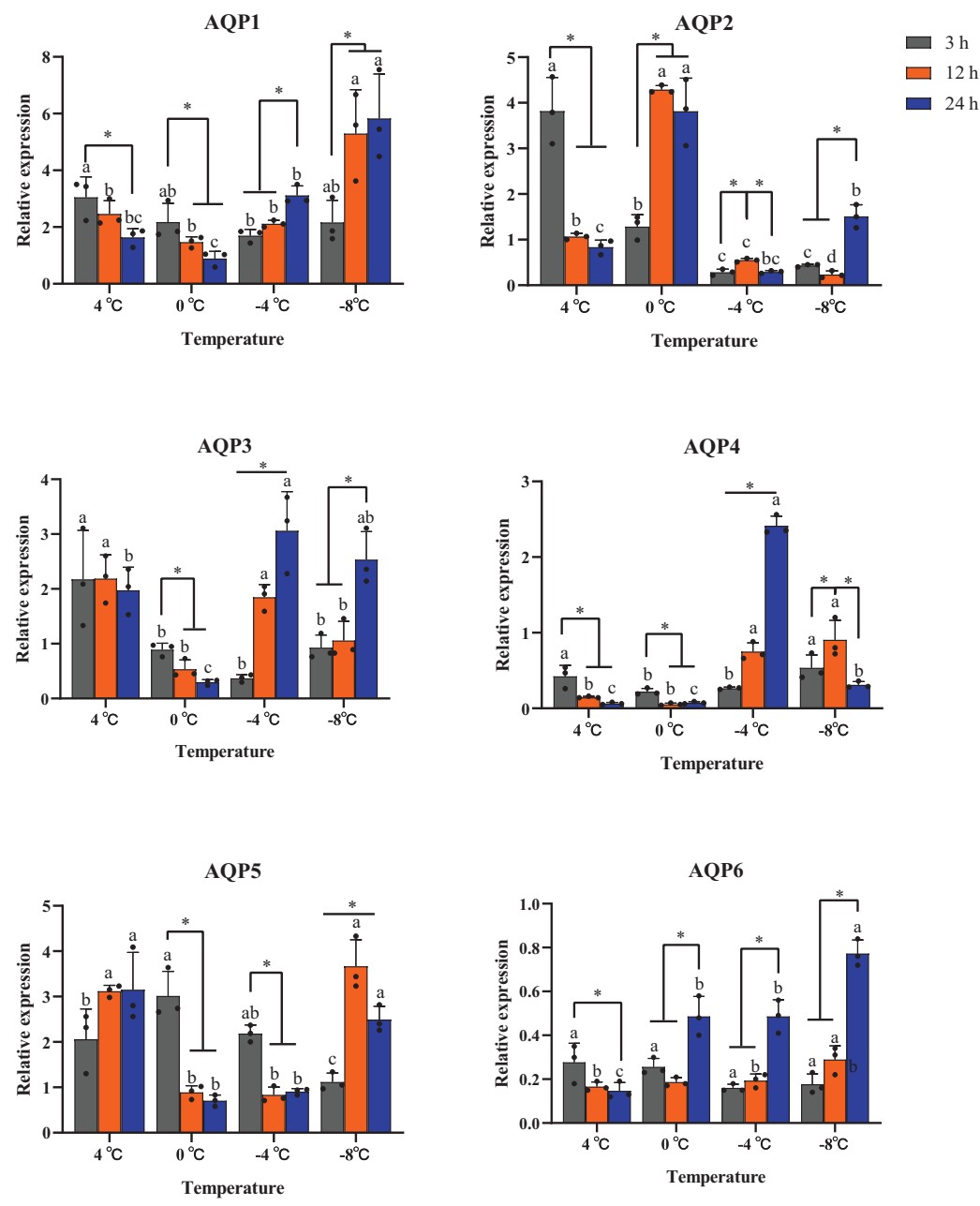

**Figure 5 Relative mRNA expression levels of *AQP* 1-6 under different duration of low temperature treatment.** The relative expression levels were normalized with *Actin* and using the expression levels at 23 °C for calibration. The standard errors of the means of three biological replicates are represented by error bars. Means separated by one-way ANOVA ($P < 0.05$). Values are denoted as the mean ± SE ($n = 3$). Significant differences between temperatures (4 °C, 0 °C, −4 °C, −8 °C) during the same cooling-time are marked with letters, and different cooling time (3 h, 12 h, 24 h) under a certain temperature are marked with an asterisk (*).                

The three-dimensional structures analysis provided an intuitive insight of the potential function of various channel permeability. Drip and Prip are water-selective, and they mainly transport water molecules (*Sui et al., 2001*; *Tan & Chen, 2023*), while Glps are permeable to small molecules such as glycerol or urea. However, their functions may be

variable due to the posttranslational modifications, the cellular environment they expressed (*Finn et al., 2015*). Our results suggested that AaAQP3 was entomoglyceroporins (Eglp), the composition of the ar/R constriction site and the corresponding larger (1.9 Å), more hydrophobic three-dimensional shape of the pore enables permeation of additional solutes. While AaAQP1 was a water-specific classical aquaporins equipped with a tighter pore which contraction region radius close to 1.5 Å, allowing water molecules (diameter = 2.8 Å) to pass through. These findings may provide the basis for the important role of AaAQPs in its larval cold tolerance.

The expression levels of several AQP genes, such as *AQP1*, *AQP*3 and *AQP*6, were significantly up-regulated in *A. aeneociliella* larvae after low temperature exposed, indicating that AQPs might be involved in its cold tolerance. *AQP1* belongs to PRIP subfamily, which plays an important role in the regulation of cellular water osmotic pressure and proposed to improve insect cold tolerance (*Campbell et al., 2008*; *Kataoka, Miyake & Azuma, 2009*; *Fu et al., 2019*). For example, the transcript levels of *MpAQP1* in *Microdera punctipennis* was up-regulated after treated at 4 °C (*Li, Liu & Ma, 2017*). Although the AQP12L subfamily was identified in recent years (*Sandhya & Kavitha, 2018*), the physiological functions of AaAQP6 in insects need to be further investigated.

The cells of cryotolerant organisms must coordinate the outflow of water and the inflow of cryoprotectants in response to the concentration of extracellular solute during freezing (*Philip, Kiss & Lee, 2011*). Glycerol, urea and trehalose are important cryoprotectants in insects, and can increase the concentration of the cells and thus reduce the SCP, or directly interact with enzymes and other proteins to play a protective role (*Tomcala et al., 2007*; *Matsukura et al., 2008*). The higher expression of Eglp clade AQPs contributes to the increase of glycerol permeability on the cell membrane, and promotes the influx of more glycerol into the cell, which enables the insects to maintain homeostasis in the low temperature environment (*Izumi et al., 2006*; *Finn et al., 2015*). Moreover, the SeAQP of beet armyworm, *Spodoptera exigua* was not only involved in the transport of glycerol under rapid cold hardening (RCH), but also associated with glycerol kinase to produce glycerol, through which the newly synthesized glycerol may accumulate in haemolymph (*Ahmed & Kim, 2019*). The potential functions of these cryoprotectants in *A. aeneociliella* need further investigations.

## CONCLUSIONS

The AQPs are key players in osmoregulation to enhance the cold resistance in many animals. However, AQP remains poorly characterized in insects. Our studies showed that six AaAQP sequences belonging to five arthropod AQP subfamilies in *A. aeneociliella* were successfully cloned. The results of deduced 3-D structure and RT-qPCR indicated that AQPs had a potentially important role in the cold tolerance of *A. aeneociliella*. AaAQP1 and AaAQP3 were probably considered to be the key anti-hypothermia proteins in *A. aeneociliella* by regulating rapid cell dehydration and allowing the influx of extracellular cold resistance molecules, thus avoiding death in the winter wheat fields.

### Funding

This study was supported by the Natural Science Foundation of Shandong Province (ZR2020MC121) and the National Key R&D Program of China (2017YFD0201705). The funders had no role in study design, data collection and analysis, decision to publish, or preparation of the manuscript.

### Grant Disclosures

The following grant information was disclosed by the authors:
Natural Science Foundation of Shandong Province: ZR2020MC121.
National Key R&D Program of China: 2017YFD0201705.

### Competing Interests

The authors declare that they have no competing interests.

### Author Contributions

- Chunqing Zhao conceived and designed the experiments, analyzed the data, authored or reviewed drafts of the article, and approved the final draft.
- Zhen Liu performed the experiments, prepared figures and/or tables, authored or reviewed drafts of the article, and approved the final draft.
- Yong Liu conceived and designed the experiments, prepared figures and/or tables, authored or reviewed drafts of the article, and approved the final draft.
- Yidi Zhan performed the experiments, analyzed the data, authored or reviewed drafts of the article, and approved the final draft.

### DNA Deposition

The following information was supplied regarding the deposition of DNA sequences:
The sequences are available at GenBank: AQP1, MW543947; AQP2, MW543948; AQP3, MW543949; AQP4, MW543950; AQP5, MW543951; AQP6, MW543952.

### Data Availability

The raw measurements are available in the Supplemental File.

### Supplemental Information

Supplemental information for this article can be found online at http://dx.doi.org/10.7717/peerj.16403#supplemental-information.

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
