# Peer review of "Identification and characterization of cold-responsive aquaporins from the larvae of a crambid pest Agriphila aeneociliella (Eversmann) (Lepidoptera: Crambidae)"

_PeerJ, doi:10.7717/peerj.16403_

## Round 0.1 · original submission · Minor Revisions

Your article has been seen by 3 expert reviewers, 2 of whom suggested minor revisions and one of whom suggested major revisions. I have labelled it minor revisions, but with conditions.

Reviewers noted incomplete references to the recent literature. This should be fixed. A number of places with unclear wording also need improvement.

Reviewer #3 had more in-depth criticisms. Their points 2-6 should be clearly addressed in the revision. Their point 1 however calls for more experiments; if their points 2-6 are fully addressed I would not ask for tissue-specific experiments. However the authors should keep this criticism in mind.

dCT measurements are indeed only as good as the reference gene used. Points 4 and 6 need to be carefully considered. Including a table of CT values for all replicates and genes will be required for publication.

**Language Note:** The review process has identified that the English language must be improved. PeerJ can provide language editing services - please contact us at copyediting@peerj.com for pricing (be sure to provide your manuscript number and title). Alternatively, you should make your own arrangements to improve the language quality and provide details in your response letter. – PeerJ Staff

Reviewer 1 ·

Basic reporting

The paper is original and is of importance to readers. In this study, the authors studied Aquaporins (AQPs) in Agriphila aeneociliella, and indicated that AQP genes were belonged to five AQP subfamilies (RPIP, Eglp, AQP12L, PRIP, DRIP). Moreover, the transcription levels of AaAQP 1, 3, and 6 increased under cold. The author suggested that AaAQP1 and AaAQP3 were probably considered to be the key anti-hypothermia proteins in A. aeneociliella by regulating cold resistance molecules. The experiments were clear and the results were well presented.

Experimental design

Materials and methods: Line 108: why you set 3 time nodes? And I know that the nodes contained short-term cold stress and long-term cold stress, they are different.

Validity of the findings

Results: Line 227-228: “Therefore, ...” , the conclusion that means AQPs in A. aeneociliella may play the similar roles in enhancing the insect resistance to cold. I don’t understand, and the family AQP genes may show diverse functions, you get the results of the Classical aquaporin and Glps, but how about AQP12L?

Additional comments

Results: Line 237: “AaAQP1, 3, and 6”, they are not italic, I suggest that change to AaAQP1, AaAQP3, and AaAQP6.
Discussion: Line 259: “Hg2+-sensitive cysteine”, you may make sure whether superscript of “Hg2+”, like Hg2+.
The reference needs to be updated. There are relatively few new references. It is the best after 2018, and at least after 2016. Please increase the proportion of recent references.

·

Basic reporting

The manuscript requires some proofreading before acceptance. Some examples include line 19: “which belonged” should be “which belong”; line 21: “Asparagine-Proline-Alanine motif” should be “Asparagine-Proline-Alanine motifs”; the title of the manuscript line 2: “cramid” should be “crambid”. I suggest having colleagues proficient in English review and edit the manuscript.

Experimental design

No comment

Validity of the findings

I commend the authors on fig. 2 containing the comparisons of AQPs to other arthropods. However, I only see a close relative to mosquitoes in clade A PRIP. The first mosquito aquaporins to be characterized were in Aedes aegypti (Drake et al., PLoS One 2010; Drake et al., Sci Rep 2015) and it was found that A. aegypti are closely related to A. gambiae. It might benefit the figure to expand and include arthropods that have been previously characterized.

Reviewer 3 ·

Basic reporting

Great paper, but it will require substantial English language revisions - respectfully, in many cases, it is completely unclear what the authors are trying to communicate, whereas in other instances the word choice sounds odd.
E.g., many instances in the abstract, l. 82, l. 263, etc

Lots of important AQP studies in insects are missing from the citation record. Need to cover the literature
better for intro and discussion… No studies cited in intro with experiments performed directly in animals, not just phylogeny of AQPs. Both intro and discussion need heavy revisions in my professional opinion.

I would also suggest proofreading the manuscript for typographical errors - ll. 39, 40, 69, 259

Experimental design

I have several substantial methodological concerns about the qPCR data:

1. The study implies whole-animal mRNA abundance analysis - to me this seems
that it may not provide the resolution the authors need to support their conclusions.
AQPs are expressed in multiple tissues and those tissues may undergo opposing
mRNA abundance changes leading to false negatives. Vice versa may also be true.
A tissue-specific analysis is needed here really.

2. l. 110 is implying a mortality bottleneck - what was the mortality like? %-wise?

3. what was the quality of the RNA like? report average A260/A280 and A260/A230 values here please

4. accession #? what isoform? of B-actin used for qPCR analysis?

5. Three biological replicates in general are not enough to represent the data as
average +/- s.e.m. -> typically anything under n=5 you’d need to plot so the reader
can see individual values as well as the average.

6. In general, when ectotherms are exposed to lower temperatures,
transcription levels go down overall. If ddCt method is used for
mRNA abundance calculations, is actin significantly changing
when the larvae arfe exposed to different temperatures?
If it is, this would disqualify it as a good housekeeping gene for qPCR
analysis in your case. Specifically, what are the Ct values of actin
in all groups? Please provide a similar statistical analysis with p-values
demonstrating that abundance of actin is not changing under different temperatures
and time points.

Validity of the findings

At the very least, proper statistics and reporting of the changes in Ct values of the housekeeping gene used will be needed to validate the qPCR results. Whole-body mRNA abundance analysis needs to be acknowledged as a limitation. In my humble opinion, whole-body analysis of AQP mRNA abundance is of very limited use for physiological conclusions.

Additional comments

Please see attached PDF for minor comments - I suggest revising for clarity, grammar and typography, then rewriting in the light of concerns raised above. I will review a revised submission ONLY if proper statistics are used and if actin Ct values are provided.

Annotated reviews are not available for download in order to protect the identity of reviewers who chose to remain anonymous.

---

## Round 0.2 · accepted · Accept

Dear authors,
Two of three reviewers recommended acceptance of your article. Since the 3rd reviewer will not be satisfied without more data, I suggest acceptance.

Reviewer 1 ·

Basic reporting

The english in article not very technically correct text. The article must improve the english expression.
The article showed less newly literature references, the author could add some new references.

Experimental design

The meaningful replication was not found in article. the experimental design was conventional. The time of treatments in the AQP genes expression levels was less and short. Under cold, the insects show long-term and short-term clod tolerance, I suggest to add more treatments or could add short-term reresponsive, because the survival of larvae during winter have long time, I advise show different cold treatment, to identify cold-responsive of larvae.

Validity of the findings

The discussion just show the resluts, but the conclusion or the meaningful replication were less.

Additional comments

The section "Expression Profiles of AaAQPs Exposed to Low Temperatures" were not show correct expression, please add "F = 22.170, P = 0.02",like this after different treatments.

·

Basic reporting

No comment.

Experimental design

No comment.

Validity of the findings

No comment.

Additional comments

No comment.

Reviewer 3 ·

Basic reporting

Dear authors,

Thank you for your revisions and for transparently providing your raw data! All of my major concerns have been addressed. I understand the whole-body limitation of the study. I have one final suggestion, but I would leave this up to the editor o whether it is requested or not (i.e., no need to send the ms back to me - whatever y'all are happy with is fine with me):

The data as presented in Fig 5 have two independent variables - time and temperature - and should really be analyzed by a two-way ANOVA. Totally up to you whether you'd like to re-analyse and re-annotate or leave as is.

Great ms!!!

Experimental design

See above

Validity of the findings

See above

Additional comments

See above